# Uncertainty Quantification via Neural Posterior Principal Components

**Elias Nehme**
Technion - Israel Institute of Technology
seliasne@campus.technion.ac.il

**Omer Yair**
Technion - Israel Institute of Technology
omeryair@campus.technion.ac.il

**Tomer Michaeli**
Technion - Israel Institute of Technology
tomer.m@ee.technion.ac.il

## Abstract

Uncertainty quantification is crucial for the deployment of image restoration models in safety-critical domains, like autonomous driving and biological imaging. To date, methods for uncertainty visualization have mainly focused on per-pixel estimates. Yet, a heatmap of per-pixel variances is typically of little practical use, as it does not capture the strong correlations between pixels. A more natural measure of uncertainty corresponds to the variances along the principal components (PCs) of the posterior distribution. Theoretically, the PCs can be computed by applying PCA on samples generated from a conditional generative model for the input image. However, this requires generating a very large number of samples at test time, which is painfully slow with the current state-of-the-art (diffusion) models. In this work, we present a method for predicting the PCs of the posterior distribution for any input image, in a single forward pass of a neural network. Our method can either wrap around a pre-trained model that was trained to minimize the mean square error (MSE), or can be trained from scratch to output both a predicted image and the posterior PCs. We showcase our method on multiple inverse problems in imaging, including denoising, inpainting, super-resolution, colorization, and biological image-to-image translation. Our method reliably conveys instance-adaptive uncertainty directions, achieving uncertainty quantification comparable with posterior samplers while being orders of magnitude faster. Code and examples are available on our webpage.

## 1 Introduction

Reliable uncertainty quantification is central to making informed decisions when using predictive models. This is especially important in domains with high stakes such as autonomous cars and biological/medical imaging, where based on visual data, the system is asked to provide predictions that could influence human life. In such domains, communicating predictive uncertainty becomes a necessity. In the particular case of image-to-image inverse problems, efficient *visualization* of predictive uncertainty is required. For example, in biological and medical image-to-image translation [7, 35, 37, 38], the predicted image as a whole is supposed to inform a scientific discovery or affect the diagnosis of a patient. Hence, an effective form of uncertainty to consider in this case is semantically-coordinated pixel variations that could alter the output image.

Currently, the majority of existing methods handle uncertainty in image-valued inverse problems by factorizing the output posterior into per-pixel marginals, in which case the strong correlations between pixels are completely ignored. This leads to per-pixel uncertainty estimates in the form

37th Conference on Neural Information Processing Systems (NeurIPS 2023).

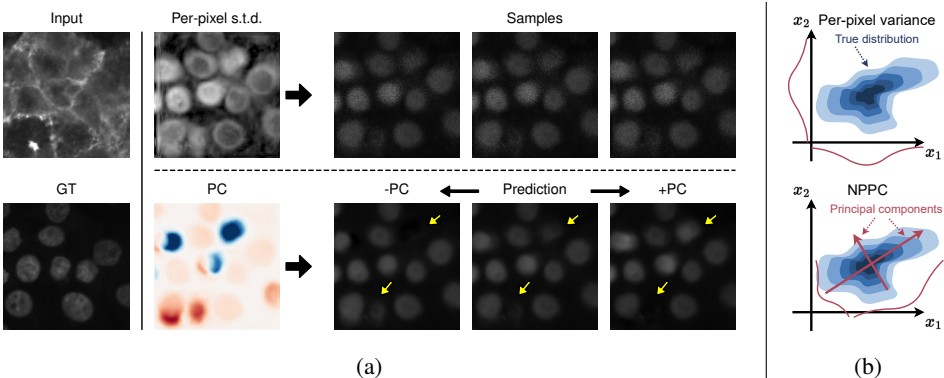

(a)                             (b)

Figure 1: **Comparison between per-pixel methods and NPPC**. In structured output posteriors such as in image-to-image regression (left), uncertainty visualization in the form of per-pixel variance maps (top) fail to convey semantic variations in the prediction, leaving the user with no clue regarding the different possibilities of the solution set. Our method - NPPC (bottom) captures a more natural measure of uncertainty, by providing the user with input-adaptive PCs around the mean prediction. These can then be used to navigate the different possibilities in a meaningful manner. *e.g.,* in 1a NPPC is used to traverse the existence of uncertain cells in an image-to-image translation task from biological imaging. Panel 1b presents a 2D motivating example illustrating the practical benefit of moving along PCs as opposed to the "standard" coordinate system (*e.g.,* pixels in an image).

of variance heatmaps [20] or confidence intervals [2], which are of little practical use as they do not describe the *joint* uncertainty of different pixels. As a result, such models suffer from model misspecification, absorbing inter-pixel covariates into per-pixel marginals, and presenting the user with unnecessarily inflated uncertainty estimates (see Fig. 1b). For example, consider a set of pixels along an edge in an image, in the task of image super-resolution. It is clear that for severe upsampling factors, a model would typically be uncertain around object edges. However, since these pixels are all correlated with the same object, the underlying uncertainty is whether the entire edge is shifted *jointly*, and not whether individual pixels can be shifted independently.

Recently, the field of image-to-image regression has seen a surge of probabilistic methods inspired by advancements in deep generative models, such as diffusion models (DMs). The recurring theme in these approaches is the use of a deep generative prior in order to sample from the posterior distribution [8, 19, 50]. In principle, using the resulting samples, it is possible to present the practitioner with the main modes of variations of the posterior uncertainty, for example using principal components analysis (PCA). However, these powerful posterior samplers come with a heavy computational price. Despite tremendous efforts over the past few years [27, 31, 43, 51], sampling from state-of-the-art methods is still unacceptably slow for many practical applications.

In this work, we bypass the need to learn a data-hungry and extremely slow conditional generative model, by directly training a neural model to predict the principal components of the posterior. Our method, which we term *neural posterior principal components* (NPPC), can wrap around any pre-trained model that was originally trained to minimize the mean square error (MSE), or alternatively can be trained to jointly predict the conditional mean alongside the posterior principal components. In particular, when used as a post-hoc method with MSE pre-trained models, NPPC is a general technique that can be transferred across datasets and model architectures seamlessly. This is because our network architecture inherits the structure and the learning hyper-parameters of the pre-trained model with only one key change: Increasing the filter count at the last layer and employing a Gram-Schmidt process to the output to ensure our found principal components are orthogonal by construction. NPPC is then trained on the pre-trained model residuals, to endow the model's point prediction with efficient input-adaptive uncertainty visualization.

Several prior works proposed modeling the posterior as a correlated Gaussian distribution. In particular, Dorta et al. [9] approximated the output precision matrix by a matrix of the form $\mathbf{LL}^T$, where $\mathbf{L}$ is a lower triangular matrix with a pre-determined sparsity pattern. More recently, Monteiro et al. [33] and Meng et al. [30] approximated the covariance matrix by a sum of a diagonal matrix and a low-rank one. However, unlike NPPC, both [9] and [30] are limited to very low-resolution

images, because either they explicitly construct the (huge) covariance matrix during training or they capture only short-range correlations. Moreover, [33] suffers from training instability as multivariate Gaussians with both unknown mean and covariance are known to be notoriously unstable [46]. This limits its applicability to image segmentation, requiring proper background masking to avoid infinite covariance and overflow errors. Our method, on the other hand, does not construct the covariance matrix at any point. It directly outputs the top eigenvectors of the covariance and trains using the objective of PCA. As a result, our method is generally applicable to any inverse problem, enjoys faster and more stable training, and can handle high-resolution images as we demonstrate in Sec. 4. This is enabled by several design choices introduced to the architecture and training, including a Gram-Schmidt output layer, a PCA loss function maximizing correlation with the residuals, and a `Stopgrad` trick that enables learning all principal directions jointly, mitigating the need for training separate models for separate PCs.

We compare our PCs to those extracted from samples from a conditional generative model as recently proposed in [4]. As we show, we obtain comparable results, but orders of magnitude faster. Finally, we showcase NPPC on multiple inverse problems in imaging showing promising results across tasks and datasets. In particular, we apply NPPC to scientific biological datasets, showing the practical benefit of our uncertainty estimates in their ability to capture output correlations.

## 2  Related Work

**Model uncertainty**  Some methods attempt to report uncertainty that is due to model misspecification and/or out-of-distribution data. Early work on quantification of such uncertainty in deep models has focused on Bayesian modeling by imposing distributions on model weights/feature activations [6, 17, 34]. These methods employ various techniques for approximating the posterior of the weights, including using MCMC [44], variational inference [6, 25], Monte-Carlo Dropout [13], and Laplace approximation [36]. Our work rather focuses on *data* uncertainty as described in Section 3.

**Per-pixel methods**  Some methods for predicting per-pixel variances assume a Gaussian distribution and learn it by maximizing the log-likelihood [20]. This approach was later combined with hierarchical priors on likelihood parameters to infer a family of distributions in a single deterministic neural network [1, 28]. Similarly, multi-modal approximations assuming a Gaussian Mixture Model per pixel have been proposed [5]. Deep Ensembles [22] and Test-time augmentations [53] have also been used to estimate uncertainty by measuring the variance of model predictions. More recently, distribution-free methods such as quantile regression and conformal prediction [39] have taken over their counterparts with firm statistical guarantees. Specifically for image-to-image regression, a work that stands out is that of Angelopoulos et al. [2]. The main shortcoming of these methods is the underlying assumption of *i.i.d.* pixels.

**Distribution-free risk control**  Elaborate distribution-free methods, such as Risk Controlling Prediction Sets (RCPS) [3], take pixel correlations into account by controlling some risk factorizing all pixels in an image (such as the false discovery rate in binary image segmentation). Recently, this approach has been deployed in image-to-image regression problems using a technique called Conformal Prediction Masks [21]. The main drawback of these methods is their inability to explore the different possible options, but rather, only impose upper and lower bounds on the possible solution set. In addition, they also require an extra data split for calibration, which is not always readily available. An interesting recent work in this field [45] has proposed to control the risk of disentangled factors in the latent space of StyleGAN, and demonstrated informative uncertainty visualizations in inverse problems of facial images. However, the main drawbacks of this work are the key assumption of access to disentangled latent representations and the generalization gap from fake to real images.

**Generative models and posterior samplers**  Generative models have been widely used to navigate prediction uncertainty, either in the form of conditional variational autoencoders [47] and conditional GANs[32], or more recently using state-of-the-art score-based and denoising diffusion models [8, 12, 14, 19, 26, 41, 42, 48, 49, 54]. While the latter have achieved astounding results in the last two years, when aiming for high-quality samples, they remain extremely slow to sample from, despite promising results reported in recent efforts [27, 31, 43, 51]. The recent Conffusion method [15] finetunes pre-trained diffusion models to output interval bounds in a single forward pass to achieve fast confidence interval prediction for posterior samplers. Nonetheless, the result is a per-pixel uncertainty map.

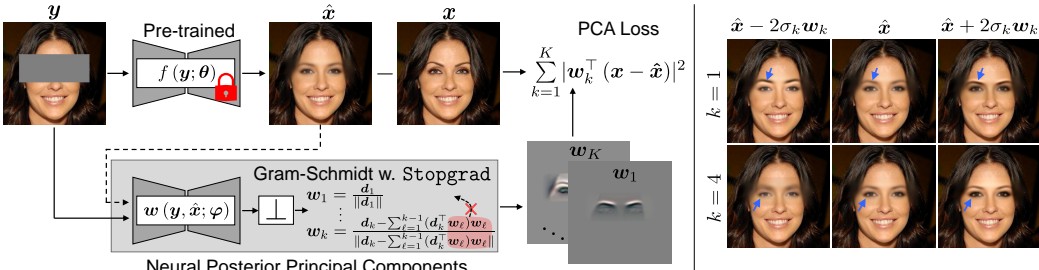

Figure 2: **Method overview**. Here, NPPC is demonstrated for image inpainting, where $x$ is the ground truth image, $y$ is the masked input, and $\hat{x}$ is the posterior mean prediction. Our method can wrap around pre-trained conditional mean predictors $f(y;\theta)$, replicating their architecture with a slight modification at the output (see text). Using a PCA loss on the errors ($e = x - \hat{x}$), we learn to predict the first $K$ PCs of the posterior $w_1, \ldots, w_K$ directly in a single forward pass of a neural network $w(y, \hat{x}; \varphi)$. On the right, we visualize the uncertainty captured by two PCs ($w_1, w_4$) around the mean prediction $\hat{x}$.

**Gaussian covariance approximation**    Works that are particularly related to the approach presented here are [9], [33] and [30]. Specifically, Dorta et al. [9] proposed to model the output posterior with a correlated Gaussian, and approximated the precision matrix using the factorization $\Lambda = \mathbf{L}\mathbf{L}^\top$, where $\mathbf{L}$ is a lower triangular matrix with predefined sparsity patterns capturing short-term correlations. The matrix $\mathbf{L}$ is then learned during training using a maximum likelihood objective. Similarly, [33] and [30] suggested approximating the posterior covariance matrix using a low-rank factorization with the latter involving second-order scores. However, the main drawback of [9] and [30] is their limited ability to generalize to high-resolution RGB images while efficiently capturing long-range correlations [10]. Additionally, [33] is limited by training instability, limiting its applicability to image segmentation. In contrast, our method directly estimates eigenvectors of the covariance matrix without the need for explicitly storing the covariance matrix in memory during training as in [30]. This enables us to seamlessly generalize to arbitrary inverse problems and high-resolution RGB images with more stable and less expensive training steps.

**Concurrent work**    A related approach was recently proposed in the concurrent work of [29], where the authors demonstrate a training-free method to calculate the posterior PCs in the task of Gaussian denoising. The key advantage of our approach over this work is that we are not constrained to the task of Gaussian denoising and can handle arbitrary inverse problems, as we show in our experiments.

## 3    Neural Posterior Principal Components

We address the problem of predicting a signal $x \in \mathbb{R}^{d_x}$ based on measurements $y \in \mathbb{R}^{d_y}$. In the context of imaging, $y$ often represents a degraded version of $x$ (*e.g.,* noisy, blurry), or a measurement of the same specimen/scene acquired by a different modality. We assume that $x$ and $y$ are realizations of random vectors $\mathbf{x}$ and $\mathbf{y}$ with an unknown joint distribution $p(x, y)$, and that we have a training set $\mathcal{D} = \{(x_i, y_i)\}_{i=1}^{N_d}$ of matched input-output pairs independently sampled from that distribution.

Many image restoration methods output a single prediction $\hat{x}$ for any given input $y$. A common choice is to aim for the posterior mean $\hat{x} = \mathbb{E}[\mathbf{x}|\mathbf{y} = y]$, which is the predictor that minimizes the MSE. However, a single prediction does not convey to the user the uncertainty in the restoration. To achieve this goal, here we propose to also output the top $K$ principal components of the posterior $p(x|y)$, *i.e.,* the top $K$ eigenvectors of the posterior covariance $\mathbb{E}[(\mathbf{x} - \hat{x})(\mathbf{x} - \hat{x})^\top|\mathbf{y} = y]$. The PCs capture the main directions along which $\mathbf{x}$ could vary, given the input $y$, and thus provide the user with valuable information. However, their direct computation is computationally infeasible in high-dimensional settings. The main challenge we face is, therefore, how to obtain the posterior PCs without having to ever store or even compute the entire posterior covariance matrix. Before we describe our method, let us recall the properties of PCA in the standard (unconditional) setting.

**Unconditional PCA**    Given a set of $N$ data points $\{x_i\}$ where $x_i \in \mathbb{R}^{d_x}$, the goal in PCA is to find a set of $K$ orthogonal principal directions $w_1, \ldots, w_K$ along which the data varies the most. The resulting directions are ordered such that the variance of $\{w_1^\top x_i\}$ is biggest, the variance of

$\{\boldsymbol{w}_2^\top \boldsymbol{x}_i\}$ is the second biggest, etc. The objective function for finding these directions has multiple equivalent forms and interpretations. The one we exploit here is the iterative maximization of variance. Specifically, let $\boldsymbol{X}$ denote the centered data matrix whose rows consist of different observations $\boldsymbol{x}_i^\top$ after subtracting their column-wise mean, and let $\boldsymbol{w}_1, \ldots, \boldsymbol{w}_K$ denote the first $K$ PCs. Then the $k^{\text{th}}$ PC is given by

$$\boldsymbol{w}_k = \arg\max_{\boldsymbol{w}} \|\boldsymbol{X}\boldsymbol{w}\|_2^2, \quad \text{s.t.} \quad \|\boldsymbol{w}\| = 1, \ \boldsymbol{w} \perp \text{span}\{\boldsymbol{w}_1, \ldots, \boldsymbol{w}_{k-1}\}, \tag{1}$$

where for the first PC, the constraint is only $\|\boldsymbol{w}\| = 1$. The variance along the $k^{\text{th}}$ PC is given by $\sigma_k^2 = \frac{1}{N}\|\boldsymbol{X}\boldsymbol{w}_k\|_2^2$.

**The challenge in posterior PCA**   Going back to our goal of computing the posterior PCs, let us assume for simplicity we are given a pre-trained conditional mean predictor $\hat{\boldsymbol{x}} = f(\boldsymbol{y}; \boldsymbol{\theta})$ obtained through MSE minimization on the training set $\mathcal{D}$. Let $\boldsymbol{e}_i = \boldsymbol{x}_i - \hat{\boldsymbol{x}}_i$ denote the error of the conditional mean predictor given the $i^{\text{th}}$ measurement, $\mathbf{y} = \boldsymbol{y}_i$. The uncertainty directions we wish to capture per sample $\boldsymbol{y}_i$, are the PCs of the error $\boldsymbol{e}_i$ around the conditional mean estimate $\hat{\boldsymbol{x}}_i$. Conceptually, this task is extremely difficult as during training for every measurement $\boldsymbol{y}_i$, we have access only to a *single* error sample $\boldsymbol{e}_i$. If we were to estimate the PCs directly, the result would be a trivial single principal direction equaling $\boldsymbol{e}_i$ which is unpredictable at test time. To address this challenge, here we propose to harness the implicit bias of neural models and to learn these directions from a dataset of triplets $\mathcal{D}' = \{(\boldsymbol{x}_i, \boldsymbol{y}_i, \hat{\boldsymbol{x}}_i)\}_{i=1}^{N_d}$. The key implicit assumption underlying our approach (and empirical risk minimization in general) is that the posterior mean $\mathbb{E}[\mathbf{x}|\mathbf{y} = \boldsymbol{y}]$ and the posterior covariance $\mathbb{E}[(\mathbf{x} - \hat{\mathbf{x}})(\mathbf{x} - \hat{\mathbf{x}})^\top | \mathbf{y} = \boldsymbol{y}]$ vary smoothly with $\boldsymbol{y}$. Hence, with the right architecture, such models can capitalize on inter-sample dependencies and properly generalize to unseen test points, by learning posterior PCs that change gracefully as a function of $\boldsymbol{y}$. This is much like models trained with MSE minimization to estimate the conditional mean $\mathbb{E}[\mathbf{x}|\mathbf{y} = \boldsymbol{y}]$, while being presented during training only with a single output $\boldsymbol{x}_i$ for every measurement $\boldsymbol{y}_i$.

### 3.1   Naive solution: Iterative learning of PCs

Following the intuition from the previous section, we can parameterize the $k^{\text{th}}$ PC of the error using a neural network $\boldsymbol{w}_k(\boldsymbol{y}, \hat{\boldsymbol{x}}; \boldsymbol{\varphi}_k)$ with parameters $\boldsymbol{\varphi}_k$, which has similar capacity to the pre-trained model $f(\boldsymbol{y}; \boldsymbol{\theta})$ outputting the conditional mean. This model accepts the measurement $\boldsymbol{y}$ and (optionally) the conditional mean estimate $\hat{\boldsymbol{x}}$, and outputs the $k^{\text{th}}$ PC of the error $\boldsymbol{e}$.

Let $\boldsymbol{d}_1(\boldsymbol{y}, \hat{\boldsymbol{x}}; \boldsymbol{\varphi}_1)$ be a model for predicting the first unnormalized direction, such that $\boldsymbol{w}_1(\boldsymbol{y}_i, \hat{\boldsymbol{x}}_i; \boldsymbol{\varphi}_1) = \boldsymbol{d}_1(\boldsymbol{y}_i, \hat{\boldsymbol{x}}_i; \boldsymbol{\varphi}_1)/\|\boldsymbol{d}_1(\boldsymbol{y}_i, \hat{\boldsymbol{x}}_i; \boldsymbol{\varphi}_1)\|$. Given a dataset of triplets $\mathcal{D}' = \{(\boldsymbol{x}_i, \boldsymbol{y}_i, \hat{\boldsymbol{x}}_i)\}$, we adopt the objective employed in (1), and propose to learn the parameters of the input-dependent first PC by minimizing

$$\mathcal{L}_{\boldsymbol{w}_1}(\mathcal{D}', \boldsymbol{\varphi}_1) = -\sum_{(\boldsymbol{x}_i, \boldsymbol{y}_i, \hat{\boldsymbol{x}}_i \in \mathcal{D}')} |\boldsymbol{w}_1(\boldsymbol{y}_i, \hat{\boldsymbol{x}}_i; \boldsymbol{\varphi}_1)^\top \boldsymbol{e}_i|^2. \tag{2}$$

Next, given the model predicting the first PC, we can train a model to predict the second PC, $\boldsymbol{w}_2(\boldsymbol{y}_i, \hat{\boldsymbol{x}}_i; \boldsymbol{\varphi}_2)$, by manipulating the output of a model $\boldsymbol{d}_2(\boldsymbol{y}_i, \hat{\boldsymbol{x}}_i; \boldsymbol{\varphi}_2)$. Specifically, following the approach in (1), we optimize the same loss (2), but construct the output of the model $\boldsymbol{w}_2$ by removing the projection of $\boldsymbol{d}_2$ onto $\boldsymbol{w}_1$, and normalizing the result. This ensures that $\boldsymbol{w}_2(\boldsymbol{y}_i, \hat{\boldsymbol{x}}_i; \boldsymbol{\varphi}_2) \perp \boldsymbol{w}_1(\boldsymbol{y}_i, \hat{\boldsymbol{x}}_i; \boldsymbol{\varphi}_1)$.

While in principle this approach can be iterated $K$ times to learn the first $K$ PCs, it has several drawbacks that make it impractical. First, it requires a prolonged iterative training of $K$ neural networks sequentially, preventing parallelization and leading to very long training times. Second, this approach is also inefficient at test time, as we need to compute $K$ dependent forward passes. Finally, in this current formulation, different PCs have their own set of weights $\{\boldsymbol{\varphi}_k\}_{k=1}^K$, and do not share parameters. This is inefficient as for a given input $\boldsymbol{y}$ and corresponding mean prediction $\hat{\boldsymbol{x}}$, it is expected that the initial feature extraction stage for predicting the different PCs, would be similar. A better strategy is therefore to design an architecture that outputs all PCs at once.

### 3.2   Joint learning of PCs

To jointly learn the first $K$ PCs of the error using a single neural network $\boldsymbol{w}(\boldsymbol{y}_i, \hat{\boldsymbol{x}}_i; \boldsymbol{\varphi})$, we introduce two key changes to the architecture inherited from the pre-trained mean estimator $f(\boldsymbol{y}; \boldsymbol{\theta})$. First,

the number of filters at the output layer is multiplied by $K$, to accomodate the $K$ PCs, $\boldsymbol{w}_1, \ldots, \boldsymbol{w}_K$. Second, we introduce a Gram-Schmidt procedure at the output layer, making the directions satisfy the orthogonality constraint by construction. Formally, denote the non-orthonormal predicted set of directions by $\boldsymbol{d}_1, \ldots, \boldsymbol{d}_K$. Then, we transform them to be an orthonormal set $\boldsymbol{w}_1, \ldots, \boldsymbol{w}_K$ as

$$
\begin{aligned}
\boldsymbol{w}_1 &= \frac{\boldsymbol{d}_1}{\|\boldsymbol{d}_1\|}, \\
\boldsymbol{w}_k &= \frac{\boldsymbol{d}_k - \sum_{\ell=1}^{k-1}(\boldsymbol{d}_k^\top \boldsymbol{w}_\ell)\boldsymbol{w}_\ell}{\|\boldsymbol{d}_k - \sum_{\ell=1}^{k-1}(\boldsymbol{d}_k^\top \boldsymbol{w}_\ell)\boldsymbol{w}_\ell\|}, \quad k = 2, \ldots K.
\end{aligned}
\tag{3}
$$

Note, however, that these changes do not yet guarantee proper learning of the PCs. Indeed, if we were to learn the directions by naively minimizing the loss

$$
\mathcal{L}_{\boldsymbol{w}}(\mathcal{D}', \boldsymbol{\varphi}) = - \sum_{(\boldsymbol{x}_i, \boldsymbol{y}_i, \hat{\boldsymbol{x}}_i \in \mathcal{D}')} \sum_{k=1}^{K} |\boldsymbol{w}_k(\boldsymbol{y}_i, \hat{\boldsymbol{x}}_i; \boldsymbol{\varphi})^\top \boldsymbol{e}_i|^2,
\tag{4}
$$

then we would only recover them up to an orthogonal matrix. To see this, let $\boldsymbol{W}_i$ denote the matrix that has $\boldsymbol{w}_k(\boldsymbol{y}_i, \hat{\boldsymbol{x}}_i; \boldsymbol{\varphi})$ in its $k^{\text{th}}$ column. Then the inner sum in (4) can be rewritten as $\|\boldsymbol{W}_i^\top \boldsymbol{e}_i\|_2^2$. Now, it is easy to see that neither the loss nor the constraints are affected by replacing each $\boldsymbol{W}_i$ with $\tilde{\boldsymbol{W}}_i = \boldsymbol{W}_i \boldsymbol{O}_i$, for some orthogonal matrix $\boldsymbol{O}_i$. Indeed, this solution satisfies the orthogonality constraint $\tilde{\boldsymbol{W}}_i^\top \tilde{\boldsymbol{W}}_i = \boldsymbol{I}$, and attains the same loss value as the original PCs.

Note that this rotation ambiguity did not exist in the naive approach of Sec. 3.1 because there, when finding the $k^{\text{th}}$ PC given the preceding $k-1$ pre-trained PCs, the loss term $|\boldsymbol{w}_k(\boldsymbol{y}_i, \hat{\boldsymbol{x}}_i; \boldsymbol{\varphi})^\top \boldsymbol{e}_i|^2$ did not affect the learning of $\boldsymbol{w}_1, \ldots, \boldsymbol{w}_{k-1}$. However, when attempting to learn all directions jointly, the preceding PCs receive a gradient signal from the $k^{\text{th}}$ loss term as $\boldsymbol{w}_k$ is a function of $\boldsymbol{w}_1, \ldots, \boldsymbol{w}_{k-1}$ in the Gram-Schmidt procedure.

To solve this problem and decouple the learning of the different PCs while still maintaining their orthogonality, we propose a simple modification to the Gram-Schmidt procedure: Using `Stopgrad` for the previously derived PCs within the projection operators in (3) (see the red term in Fig. 2). This way, in each learning step the different PCs are guaranteed to be orthogonal, while solely optimizing their respective objective. This allows learning them jointly and recovering the solution of the iterative scheme in a single forward pass of a neural network with shared parameters. Please see App. D.2 for validation of the equivalence between sequential and joint PC learning.

## 3.3 Learning variances along PCs

Recall that the variance along the $k^{\text{th}}$ direction corresponds to the average squared projection of the data along that direction. We can use that to output a prediction of the variances $\{\sigma_k^2\}$ of the PCs, by using a loss that minimizes $\sum_i(\sigma_k^2 - |\boldsymbol{w}_k^\top \boldsymbol{e}_i|^2)^2$ for every $k$. However, instead of adding $K$ additional outputs to the architecture, we can encode the variances in the norms of the unnomarlized directions. To achieve this without altering the optimization objective for finding the $k^{\text{th}}$ PC, we again invoke the `Stopgrad` operator on the term $|\boldsymbol{w}_k^\top \boldsymbol{e}_i|$ which is the current loss function value, and match the norm of the found PCs at the current step to this projected variance by minimizing the loss

$$
\mathcal{L}_{\sigma}(\mathcal{D}', \boldsymbol{\varphi}) = \sum_{(\boldsymbol{x}_i, \boldsymbol{y}_i, \hat{\boldsymbol{x}}_i \in \mathcal{D}')} \sum_{k=1}^{K} \left( \left\| \boldsymbol{d}_k - \sum_{\ell=1}^{k-1}(\boldsymbol{d}_k^\top \boldsymbol{w}_\ell)\boldsymbol{w}_\ell \right\|_2^2 - |\boldsymbol{w}_k^\top \boldsymbol{e}_i|^2 \right)^2.
\tag{5}
$$

With this, our prediction for $\sigma_k^2$ at test-time is simply $\|\boldsymbol{d}_k - \sum_{\ell=1}^{k-1}(\boldsymbol{d}_k^\top \boldsymbol{w}_\ell)\boldsymbol{w}_\ell\|^2$. Please see App. D.3 for quantitative validation of our estimated variances.

## 3.4 Joint prediction of posterior mean

Thus far, we assumed we have access to a pre-trained posterior mean predictor $\hat{\boldsymbol{x}} = f(\boldsymbol{y}; \boldsymbol{\theta})$, obtained through MSE minimization on a dataset $\mathcal{D} = \{(\boldsymbol{x}_i, \boldsymbol{y}_i)\}_{i=1}^{N_d}$ of matched pairs. Hence, our entire derivation revolved around a dataset of triplets $\mathcal{D}' = \{(\boldsymbol{x}_i, \boldsymbol{y}_i, \hat{\boldsymbol{x}}_i)\}_{i=1}^{N_d}$. However, this is not strictly necessary. In particular, the posterior mean predictor can be learned jointly alongside the PCs with an

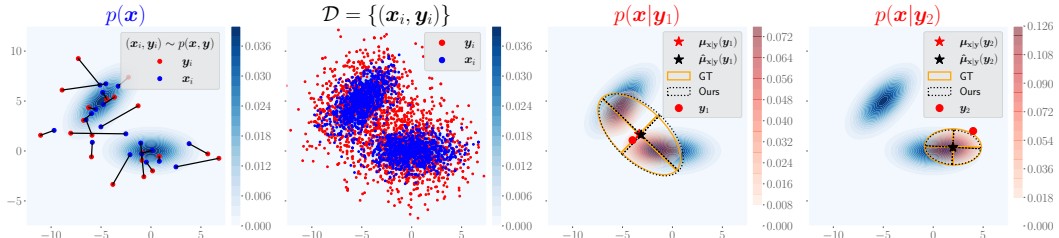

Figure 3: **Denoising samples from a 2D Gaussian mixture.** The left two panels depict the underlying (unknown) signal prior $p(\boldsymbol{x})$ (blue heatmap), exemplar matched samples from the joint distribution $(\boldsymbol{x}_i, \boldsymbol{y}_i) \sim p(\boldsymbol{x}, \boldsymbol{y})$, and the resulting training set $\mathcal{D}$. The right two panels show the analytical posterior $p(\boldsymbol{x}|\boldsymbol{y})$ (red heatmap) for two different test points $\boldsymbol{y}_1$ and $\boldsymbol{y}_2$ (marked as red dots) on top of the signal prior. Our estimated conditional mean $\hat{\boldsymbol{\mu}}_{\mathbf{x}|\mathbf{y}}(\boldsymbol{y})$ (black star) and PCs scaled by the estimated std (dashed black ellipse) coincide with the analytical posterior mean $\boldsymbol{\mu}_{\mathbf{x}|\mathbf{y}}(\boldsymbol{y})$ (red star), and scaled posterior PCs derived from the analytical covariance (solid orange ellipse).

additional MSE loss

$$\mathcal{L}_{\boldsymbol{\mu}}(\mathcal{D}, \boldsymbol{\varphi}) = \sum_{(\boldsymbol{x}_i, \boldsymbol{y}_i \in \mathcal{D})} \|\boldsymbol{x}_i - \hat{\boldsymbol{x}}_i\|_2^2. \tag{6}$$

Note that we do not want the gradients of $\mathcal{L}_{\boldsymbol{w}}$ and $\mathcal{L}_{\sigma}$ to affect the recovered mean. Therefore, we use a `Stopgrad` operator when inputting the error $\boldsymbol{x}_i - \hat{\boldsymbol{x}}_i$ to those loss functions.

### 3.5 Putting it all together

To summarize, given a dataset $\mathcal{D} = \{(\boldsymbol{x}_i, \boldsymbol{y}_i)\}_{i=1}^{N_d}$ of matched pairs, we can learn a model that predicts the posterior mean as well as the posterior PCs, using the combined loss functions of eqs. (4)-(6),

$$\mathcal{L}_{\text{all}} = \mathcal{L}_{\boldsymbol{\mu}} + \lambda_1 \mathcal{L}_{\boldsymbol{w}} + \lambda_2 \mathcal{L}_{\sigma}, \tag{7}$$

where $\lambda_1$ and $\lambda_2$ are weights balancing the contributions of the different losses.

## 4 Experiments

We now illustrate NPPC on several tasks and datasets. In all experiments except for the toy example, we used variants of the U-Net architecture [11, 40]. The weighting factors for the losses were chosen such that all three terms are roughly within an order of magnitude of each other. Empirically, we find that ramping up the weight factors for the terms of the directions and the variances after the mean estimate started converging, stabilizes training. Full details regarding the architectures, the scheduler, and the per-task setting of $\lambda_1, \lambda_2$ are in App. A.

**Toy examples** Figure 3 demonstrates NPPC on a 2D denoising task, where samples $\boldsymbol{x}_i$ from a two-component Gaussian mixture model are contaminated by additive white Gaussian noise to result in noisy measurements $\boldsymbol{y}_i$. To predict $\mathbf{x}$ from $\mathbf{y}$, we trained a 5-layer MLP with 256 hidden features using MSE minimization, and around the estimated conditional mean $\hat{\boldsymbol{x}} = \hat{\boldsymbol{\mu}}_{\mathbf{x}|\mathbf{y}}(\boldsymbol{y})$, we trained NPPC (instantiated with a similar MLP) to output the two PCs

Table 1: Comparison of the Wasserstein 2-distance from the rank $K$ Gaussian approximation of the GT posterior $p(\boldsymbol{x}|\boldsymbol{y}) \approx \mathcal{N}(\boldsymbol{x}; \boldsymbol{\mu}_{\mathbf{x}|\mathbf{y}}, \boldsymbol{\Sigma}_{\mathbf{x}|\mathbf{y}})$ (*i.e.*, $K = \text{rank}(\boldsymbol{\Sigma}_{\mathbf{x}|\mathbf{y}})$), on 5000 test samples (see text).

|  | $K=0$ | $K=3$ | $K=6$ | $K=9$ | $K=12$ |
|---|---|---|---|---|---|
| Baseline | 0.4 | 180.6 | 272.5 | 317.5 | 325.7 |
| NPPC | 0.4 | **30.3** | **31.2** | **26.6** | **28.8** |

$\boldsymbol{w}_1, \boldsymbol{w}_2$, and their variances $\hat{\sigma}_1^2, \hat{\sigma}_2^2$. As can be seen in the right two panels of Fig. 3, NPPC accurately predicts the ground-truth (GT) PCs (those are computed from the analytical expression for the posterior covariance; see App. B). To quantify the accuracy of the recovered PCs, we computed the Wasserstein 2-distance between a Gaussian approximation of the GT posterior $p(\boldsymbol{x}|\boldsymbol{y}) \approx \mathcal{N}(\boldsymbol{\mu}_{\mathbf{x}|\mathbf{y}}, \boldsymbol{\Sigma}_{\mathbf{x}|\mathbf{y}})$, and an estimated Gaussian constructed by NPPC, $\hat{p}(\boldsymbol{x}|\boldsymbol{y}) = \mathcal{N}(\hat{\boldsymbol{\mu}}_{\mathbf{x}|\mathbf{y}}, \boldsymbol{W}_\star \boldsymbol{W}_\star^\top)$, where $\boldsymbol{W}_\star$ has the scaled estimated PC $\hat{\sigma}_k \boldsymbol{w}_k$ in its $k$th column. Compared to a baseline of a point mass at the estimated

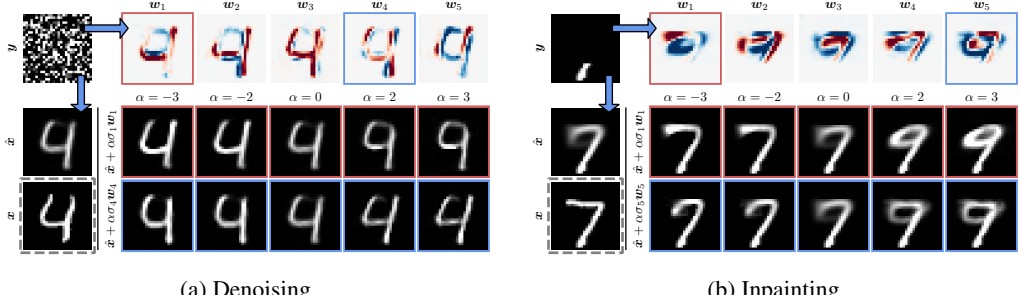

|     |     |
|-----|-----|
| (a) Denoising | (b) Inpainting |

Figure 4: **MNIST denoising and inpainting.** (a) Here we show the application of NPPC to extreme image denoising. On the left is the noisy measurement $\boldsymbol{y}$, the estimated conditional mean $\hat{\boldsymbol{x}}$, and the (unknown) ground truth test image $\boldsymbol{x}$. On the right, the top row shows the first $K = 5$ predicted PCs $\boldsymbol{w}_k$, and the bottom rows show a traversal of 3 standard deviations around $\hat{\boldsymbol{x}}$ for $\boldsymbol{w}_1, \boldsymbol{w}_4$. At an extreme noise level of $\sigma_\varepsilon = 1$, the digit is either a "4" or a "9". (b) Here we show the result of NPPC on image inpainting from only the 8 bottom rows. The PCs reveal the digit is either a "7" or a "9".

conditional mean, $\delta(\mathbf{x} - \hat{\mathbf{x}})$, NPPC reduces the Wasserstein 2-distance by $100\times$ from 4.05 to 0.04. Similarly, we also applied NPPC to a 100-dimensional Gaussian mixture denoising task, and computed the Wasserstein 2-distance from a Gaussian distribution whose mean is the GT posterior mean and whose covariance is the best rank-$K$ approximation of the GT posterior covariance. As clearly evident in Table 1, NPPC maintains a roughly constant distance to the analytical posterior, while the point mass baseline distance rapidly grows with $K$. See App. B for more details.

**Handwritten digits**  Figure 4 demonstrates NPPC on denoising and inpainting of handwritten digits from the MNIST dataset. In the denoising task, we used noise of standard deviation $\sigma_\varepsilon = 1$, and in inpainting we used a mask that covers the top 70% of the image. As can be seen, for denoising, the learned PCs capture both inter-digit and intra-digit variations, *e.g.,* turning a "4" into a "9". Similarly, in inpainting, the learned PCs traverse the two likely modes around the mean estimate $\hat{\boldsymbol{x}}$, going from a "7" into a "9". More examples are available in App. D.

**Faces**  To test NPPC on faces, we trained on the CelebA-HQ dataset using the original split inherited from celebA [24], resulting in 24183 images for training, 2993 images for validation, and 2824 images for testing. Figure 5 presents results on $256 \times 256$ face images from the CelebA-HQ dataset. Here, we demonstrate NPPC on the task of inpainting, as well as on noisy $8\times$ super-resolution with a box downsampling filter and a noise-level of $\sigma_\varepsilon = 0.05$. As can be seen, the PCs generated by NPPC capture semantically meaningful uncertainty, corresponding to the eyes, mouth, and eyebrows. We also tested our approach on inpainting of the eyes area, on $4\times$ super-resolution (noisy and noiseless), and on image colorization. More results and examples are provided in App. D.

**Comparison to posterior samplers**  While traversal along the predicted PCs is valuable for qualitative analysis, an important question is how our single forward pass PCs fair against state-of-the-art posterior samplers. To test this, we now compare NPPC to recent posterior samplers on CelebA-HQ $256 \times 256$. Specifically, we compare NPPC to DDRM [19], DDNM [54], RePaint [26], and MAT [23] on the tasks of image super-resolution/inpainting. We use each of these methods to generate 100 samples per test image, and compute PCA on those samples. We perform comparisons over 100 test images randomly sampled from the FFHQ dataset [18]. Note that NPPC and MAT were trained on the training images of CelebA-HQ according to the original split of CelebA, whereas DDRM, DDNM, and RePaint rely on DDPM [14] as a generative prior, which was trained on the entirety of CelebA-HQ. The results are reported in Table 2. As can be seen, NPPC achieves similar root MSE (RMSE) $\|\boldsymbol{x} - \hat{\boldsymbol{x}}\|_2$, and residual error magnitude $\|\boldsymbol{e} - \boldsymbol{W}\boldsymbol{W}^\top \boldsymbol{e}\|_2$ using the first $K = 5$ PCs. This is while being $100\times$ faster than MAT, and $10^3 - 10^5\times$ faster than diffusion-based methods. It is interesting to note that the principal angle between the subspace computed by NPPC and that computed by the baselines, is typically $\approx 90°$. Namely, they are nearly orthogonal. This can be attributed to the severe ill-posedness of the tested settings, resulting in multiple different PCs with roughly the same projected variance. This is also evident in the residual error magnitude, suggesting that a very large number of PCs is required to handle such extreme degradations (see App. C).

**Biological image-to-image translation**  image-to-image translation refers to translating images from one domain into another [16]. In biological imaging, image-to-image translation has been

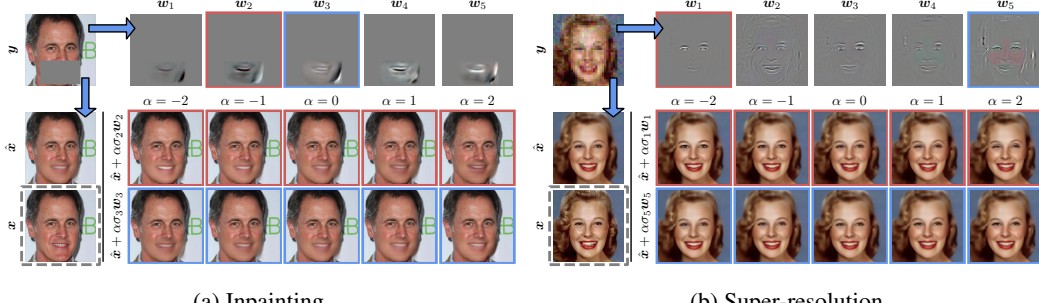

|(a) Inpainting|(b) Super-resolution|

Figure 5: **CelebA-HQ inpainting and $8\times$ noisy super-resolution**. (a) Application of NPPC to image inpainting with a mask around the mouth. NPPC received the masked image $\boldsymbol{y}$ and the mean prediction $\hat{\boldsymbol{x}}$, and predicted the first 5 posterior PCs $\boldsymbol{w}_1, \ldots, \boldsymbol{w}_5$. On the bottom, we show a traversal of 2 standard deviations around the mean estimate $\hat{\boldsymbol{x}}$ for $\boldsymbol{w}_2$ and $\boldsymbol{w}_3$, capturing uncertainty regarding open/closed mouth, and background vs shadow next to the jawline. (b) Application of NPPC to $8\times$ noisy super-resolution with a noise standard deviation $\sigma_\varepsilon = 0.05$. Similarly, the PC traversals on the bottom capture uncertainty of eye size and cheek color/position.

Table 2: Quantitative comparison of $\|\boldsymbol{x} - \hat{\boldsymbol{x}}\|_2 \downarrow$ / $\|\boldsymbol{e} - \boldsymbol{W}\boldsymbol{W}^\top \boldsymbol{e}\|_2 \downarrow$ with posterior samplers on 100 test images from FFHQ. Mean prediction and PCs were computed using 100 samples per test image, and compute is reported in neural function evaluations (NFEs).

| | Super-resolution | | | | Inpainting | | |
|---|---|---|---|---|---|---|---|
| | $4\times$ noiseless | $4\times$ noisy | $8\times$ noiseless | $8\times$ noisy | Eyes | Mouth | NFEs↓ |
| DDRM [19] | 8.94/8.89 | 11.29/11.25 | 13.68/13.47 | 16.20/16.00 | -/- | -/- | $2\cdot10^3$ |
| DDNM [54] | 8.43/8.38 | 10.74/10.70 | 13.12/12.95 | 15.73/15.55 | 13.27/11.24 | 13.30/10.37 | $10\cdot10^3$ |
| RePaint [26] | -/- | -/- | -/- | -/- | **12.24/10.3** | 12.55/**9.72** | $457\cdot10^3$ |
| MAT [23] | -/- | -/- | -/- | -/- | 14.12/12.94 | 13.08/11.74 | 100 |
| NPPC (Ours) | **8.4/8.24** | **10.41/10.35** | **13.06/12.87** | **15.29/15.11** | 13.55/11.43 | **12.23**/10.27 | **1** |

applied in several contexts, including for predicting fluorescent labels from bright-field images [35] and for predicting one fluorescent label (*e.g.,* nuclear stainings) from another (*e.g.,* actin stainings) [52]. However, unlike its use for artistic purposes, the use of image-to-image translation in biological imaging requires caution. Specifically, without proper uncertainty quantification, predicting cell nuclei from other fluorescent labels could lead to biased conclusions, as cell counting and tracking play central roles in microscopy-based scientific experiments (*e.g.,* drug testing). Here, we applied NPPC to a dataset of migrating cells imaged live for 14h (1 picture every 10min) using a spinning-disk microscope [52]. The dataset consisted of 1753 image pairs of resolution $1024 \times 1024$, out of which 1748 were used for training, and 5 were used for testing following the original split by the authors. We started by training a standard U-Net [11, 40] model using MSE minimization to predict nuclear stainings from actin stainings (see Fig. 6), and then trained NPPC on the residuals. As we show in Fig. 6, the PCs learned by NPPC convey important information to experimenters. For example, the first PC highlights the fact that accurate intensity estimation is not possible in this task, and thereby a global bias is a valid uncertainty component. Furthermore, the remaining PCs reflect semantic uncertainty by adding/removing cells from the conditional mean estimate $\hat{\boldsymbol{x}}$, thereby clearly communicating reconstruction ambiguity to the user.

## 5 Discussion

We proposed an approach for directly predicting posterior PCs and showed its applicability across multiple tasks and datasets. Nonetheless, our method does not come without limitations. First, as evident by both the PCs of NPPC and of posterior samplers, for severely ill-posed inverse problems a linear subspace with a small number of PCs captures very little of the error. Hence, a large number of PCs is required to faithfully reconstruct the error. However, the main premise of this work was scientific imaging where scientists usually take reliable measurements, and the uncertainty is not as severe as ultimately the result should drive scientific discovery. Second, throughout this paper, we

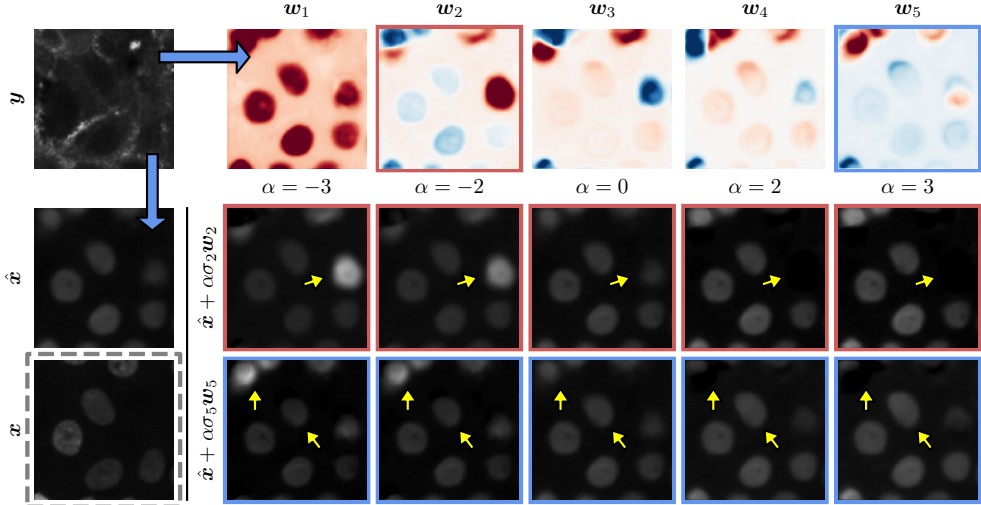

Figure 6: **Biological image-to-image translation**. NPPC applied to the task of translating one fluorescent label $y$ (actin staining) to another fluorescent label $x$ (nuclear staining) in migrating cells (left). On the right, we show the predicted PCs at the top, and traversals along the rows. Yellow arrows point to captured uncertain cells in $\hat{x}$ (either deforming or being completely erased).

learned the PCs on the same training set of the conditional mean estimator. However, generalization error between training and validation could lead to biased PCs estimation as the learning process is usually stopped after some amount of overfitting (*e.g.,* until the validation loss stagnates). This can be solved by using an additional data split for "calibration" as is typically the case with distribution-free methods [2, 3, 39, 45]. Third, different inputs $y_i$ may have posteriors of different complexities, and may thus require a different number of PCs $K$. In our approach, $K$ is hard-coded within the network's architecture (it is the number of network outputs). Namely, we treat $K$ as a hyper-parameter that needs to be set in advance prior to training, acting as an upper bound on the number of directions the user may be interested in exploring at test time. However, recall that NPPC also predicts the standard deviations along each of the $K$ PCs. These may serve to decide whether to ignore some of the PCs. Fourth, our learned PCs cover the entire image, whereas in some cases the interesting uncertainty structure could be local (*e.g.,* cell data). In such circumstances, NPPC should either be applied patch-wise or the number of PCs $K$ should be sufficiently increased. Finally, our method is tailored towards capturing uncertainty and not sampling from the posterior. While we provide the directed standard deviations, shifting along a certain direction does not guarantee samples along the way to be on the image data manifold. This can be tackled by employing NPPC in the latent space of a powerful encoder, in which case small enough steps could lead to changes on the manifold in the output image. However, this is beyond the scope of this current work.

## 6  Conclusion

To conclude, in this work we proposed a technique for directly estimating the principal components of the posterior distribution using a single forward pass in a neural network. We discussed key design choices, including a Gram-Schmidt procedure with a `Stopgrad` operator, and a principled PCA loss function. Through extensive experiments, we validated NPPC across tasks and domains, showing its wide applicability. We showed that NPPC achieves comparable uncertainty quantification to the naive approach of applying PCA on samples generated by posterior samplers while being orders of magnitude faster. Finally, we applied NPPC to the challenging task of biological image-to-image translation, demonstrating its practical benefit for safety-critical applications. In terms of broader impact, proper uncertainty quantification is crucial for trustworthy interpretable systems, particularly in healthcare applications. Thus, a method for reporting and conveniently visualizing prediction uncertainty could support users and help them avoid making flawed decisions.

## Acknowledgements

This research was partially supported by the Israel Science Foundation (grant no. 2318/22), by the Ollendorff Minerva Center, ECE faculty, Technion, and by a gift from KLA. The authors gratefully acknowledge fruitful discussions with Daniel Freedman, Regev Cohen, and Rotem Mulayoff throughout this work. The authors also thank Matan Kleiner, Hila Manor, and Noa Cohen for their help with the figures.

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
