# OpenReview forum: "Uncertainty Quantification via Neural Posterior Principal Components"
_NeurIPS.cc/2023/Conference — NeurIPS 2023 poster_

### Official Review · Reviewer_Ch4c · 2023-06-15

**Soundness:** 3 good
**Presentation:** 3 good
**Contribution:** 2 fair
**Rating:** 4
**Confidence:** 4

**Summary:**

This work proposes to model the full covariance matrix of aleatoric uncertainty for per-pixel image restoration tasks using a low-rank approximation. The low-rank approximation is estimated by a separate network trained end-to-end with the original predictor. The PCs, which are usually computed sequentially, are constructed in parallel using stop_gradiet operations.

**Strengths:**

The approach is fast and, as far as I know, architecture agnostic.

**Weaknesses:**

- The approach requires training a separate model.

- It is unclear how good parallel end-to-end training of the PCs is compared to sequential training. I could imagine that this type of approach yields pretty unstable training dynamics.

- The covariance measured by this approach is over the entire image. I could image that for pixel-wise tasks there exists at least some location invariance.

- Some important related work on low-rank approximations for pixel-wise aleatoric uncertainty [1] and efficient uncertainty estimation is missing (see works discussed in [2])

- Minor: in eq (2) w_i is normalized to unit length whereas e_i is not. Is this intended?

[1] Monteiro, M., Le Folgoc, L., Coelho de Castro, D., Pawlowski, N., Marques, B., Kamnitsas, K., van der Wilk, M. and Glocker, B., 2020. Stochastic segmentation networks: Modelling spatially correlated aleatoric uncertainty. Advances in Neural Information Processing Systems, 33, pp.12756-12767.

[2] Postels, J., Segu, M., Sun, T., Sieber, L.D., Van Gool, L., Yu, F. and Tombari, F., 2022. On the practicality of deterministic epistemic uncertainty. In Proceedings of the 39th International Conference on Machine Learning (Vol. 162, pp. 17870-17909). PMLR.

**Questions:**

- What speaks against directly parameterizing the conditional output distribution using a multivariate gaussian with low-rank approximated covariance? As far as I know, you would also never have to explicitly evaluate the cov.

- Could the authors comment on the training dynamics of the approach? How does the result compare to sequential training?

- Could the authors comment on location invariance (see weaknesses)?

- In l 160 mention that the error prediction model requires the right biases to work well. Can the authors elaborate on this?

---

> ### Author Rebuttal · Authors · 2023-08-09
>
> **Method proposition**
>
> Our work proposes to output the top $K$ covariance eigenvectors/PCs directly **without** assuming a low-rank (LR) structure. Although we can use the resulting PCs $\\bf{W}$ to construct an LR covariance approx., at no point do we use this assumption. Our work is inspired by PCA which is a non-probabilistic technique, unlike factor analysis (FA)/probabilistic PCA (PPCA) which explicitly assumes the posterior is Gaussian and approximates the full covariance matrix with $\\bf{\\Sigma}\\approx \\bf{\\Psi} +  \\bf{W}\bf{W}^{\\top}$ where $\\bf{\\Psi}$ is diagonal. In general, the 2 methods are distinct; the subspaces of PCA and FA are different unless $\\bf{\\Psi}=\\sigma^2\\bf{I}$ as in PPCA.
>
> **Missing related work**
>
> Thanks for bringing these refs to our attention. Ref [1*] is indeed related to Gaussian covariance approximations and we'll cite it in our final paper. Please note however that we did cite [25] that came out a year later in NeurIPS 21. This paper employs the same covariance approx. for image denoising rather than segmentation (a task more relevant to our work). As for ref [2*], this paper is focused on epistemic and not aleatoric uncertainty which is the focus of our work. The works dealing with aleatoric uncertainty in [2] are either referenced in our paper or focused on classification.
>
> **Why not parameterize the output using a Multivariate Gaussian (MVG) with low-rank approximated covariance?**
>
> The short answer is training stability. More specifically, the only work we are aware of using the LR MVG in image restoration is ref [25]. In principle, as you suggest, one can use this approx. in conjunction with a log-likelihood loss to impose an LR MVG on the output. In this case, the loss can be indeed calculated efficiently without explicitly evaluating the full covariance thanks to the Woodbury matrix identity and the matrix determinant lemma. However, MVGs with both unknown mean and covariance are known to be notoriously unstable to train (*e.g.* see https://arxiv.org/abs/1906.03260). In fact, as we noted in Appendix A (l 66-74, Figure A3), we found it numerically unstable to even train a per-pixel Gaussian distribution using the standard loss (eq. (A2)) and consequently resorted to eq. (A1) (similar to ref [2]). Please note that also in your ref [1*] the authors reported stability issues (paragraph “caveat” on pg. 5), and resorted to early stopping before overflow errors occurred. In addition, they also encountered infinite covariance issues in background areas and addressed it using masking. Therefore, this approach is limited by training stability, especially in the case of model misspecification. In contrast, NPPC outputs the top PCs of the covariance without imposing any probabilistic assumptions on the output distribution.
>
> **The approach requires training a separate model.**
>
> Please note that our approach enables training both the image prediction (*i.e.* the posterior mean) and the PCs jointly using a **single** model (see sections 3.4-3.5, eq. (7)), without breaking the task into two parts. However, as acknowledged in Appendix A (l 39-57, Figure A2), jointly learning the mean and the PCs led to less stable training and required more parameters+time to converge. Hence, for our CelebA-HQ/biological experiments, we focused on a two-step setting where the mean is pre-trained, and afterward, the result is wrapped around with NPPC predicting only the PCs and the variances. Please note that this limitation is not unique to NPPC, and is a standard strategy to stabilize variance prediction networks also for MVGs, *e.g.* as done in ref [6] (see also https://arxiv.org/abs/1906.03260). In fact, a similar strategy was adopted in ref [1*] that you pointed out, where even for the toy problem the authors reported stability issues and resorted to mean pre-training.
>
> **Parallel vs sequential training, and stability of training dynamics**
>
> In our experiments, when pre-training the mean we did not experience training instability. As for joint vs sequential training, this is a very good question that warrants empirical validation. To realize sequential training without significantly altering the number of parameters used to predict the PCs, we trained multiple models with an increasing number of PCs $K=1,2,\\dots,5$, and compared the PCs backward across different models. The results for the task of image colorization on CelebA-HQ confirm that end-to-end training leads to approximately the same PCs as sequential training (See Figs. R1/R2 in the rebuttal PDF). Thanks for pointing this out, we will include these results in the final version.
>
> **Location invariance**
>
> Indeed, for certain tasks, there might be some location invariance. However, please note that even if the covariance structure is local (*e.g.* a Toeplitz matrix), the PCs are not necessarily localized. In our experiments, we used a fully conv. U-net, and hence our predicted PCs were equivariant to shifts in the input (*i.e.* shifted inputs -> shifted PCs). In fact, as we explained in Appendix A (l 23-27), for the biological data we learned the PCs on $64\\times 64$ patches cropped from the full images, as cell information tends to be local. At test time, we tested on $2\\times$ bigger patches as shown in Fig. 5 and Appendix C. Ultimately, in such cases NPPC should be either applied patch-wise or the number of PCs $K$ should be increased. We will include this point in our discussion.
>
> **In eq (2) $\\bf{w}\_i$ is normalized to unit length whereas $\\bf{e}\_i$ is not. Is this intended?**
>
> As explained in Appendix A (l 58-65), we divide by $\\|\\bf{e}\_i\\|\_2\^2$ to standardize the loss values across tasks and save expensive hyperparameter tuning. We will clarify this in the text.
>
> **Biases required for NPPC to work well (l 160)**
>
> The “biases” we referred to in the text are the standard inductive/implicit biases underlying common architectures (*e.g.* convs. for images, etc). Thanks, we will clarify it.

---

### Official Review · Reviewer_2qC5 · 2023-06-16

**Soundness:** 4 excellent
**Presentation:** 4 excellent
**Contribution:** 3 good
**Rating:** 7
**Confidence:** 5

**Summary:**

For uncertainty quantification in image recovery problems, the authors propose a way to train a neural network to produce estimates of the principle components of the posterior covariance matrix. Their approach starts with a neural network that produces the posterior mean and trains a new network (or a new prediction head on the posterior-mean network) to learn the principle components and their corresponding eigenvalues. They demonstrate their method on denoising, inpainting, super-resolution, and biological image-to-image translation problems.  They show that their method produces principle component estimates of a similar quality to those of diffusion models but thousands of times faster.

**Strengths:**

1. For image recovery, the idea of visualizing posterior uncertainty using principle components is a good one and, to my knowledge, has not been adequately explored in the literature.  The most common way to present image uncertainty information is to plot a pixel-wise variance map, but this does not show pixel dependencies, which are critical to understanding the structure of the uncertainty.  And trying to visually interpret uncertainty structure from dozens of posterior image samples is tedious and heuristic.

2. The proposed method is fast at inference, since it uses only a single pass through a neural network.  This stands in contrast to Langevin/score/diffusion methods, which require generating many posterior samples, each of which can require thousands of passes through a neural network.

3. The proposed method is widely applicable because it can be built on top of an existing conditional-mean (i.e., MMSE) estimation network, which are widely available.  This stands in contrast to, say, conditional normalizing flows, whose architectural constraints make them difficult to apply/tune on new applications.

4. The numerical results are impressive.  The authors have tested their method on a wide range of applications, some of which involve consider large images (CelebA-HQ).  Also, their experiments suggest that their method gives similar RMSE and residual error magnitude to recent diffusion models.

5. The paper is very clearly written.

**Weaknesses:**

1. The authors focus on a comparison to diffusion methods, which are very slow.  But modern conditional GANs and conditional normalizing flows (CNFs) can quickly generate posterior samples with performance that meets or exceeds that of recent diffusion models.  For example, a 2022 CVPR super-resolution contest (https://arxiv.org/abs/2205.05675) showed CNFs dominating other methods.  Compared to modern CNFs (e.g., https://arxiv.org/abs/2006.14200) or CGANs (e.g., https://arxiv.org/abs/2210.13389), it’s not clear that the proposed method has any speed or performance advantages.

2. The proposed method is a one-trick pony, in that it generates only posterior principle components, and not high-quality image recoveries like posterior sampling methods do (e.g., diffusion, CGAN, CNF, etc.).  For example, the generated "x+w" face images are very blurry.  This weakness is acknowledged by the authors in Section 5.

3. The proposed method seems practical for recovering only a few principle components (e.g., tens at most), whereas in some applications the uncertainty is not well described by only a few principle components.  In other words, sometimes the eigenvalues of the posterior covariance decays very slowly.  The authors acknowledge this issue in their face experiment.  The reviewer has observed, in their own work, that slow eigenvalue decay is the rule rather than the exception.

**Questions:**

The paper was very clear and so I have no questions.

**Limitations:**

Yes

---

> ### Author Rebuttal · Authors · 2023-08-09
>
> **Advantages of the proposed method compared to fast posterior samplers such as conditional GANs/Normalizing Flows.**
>
> Thanks, this is an important point. Please note that even in the case of a fast conditional generative model capable of sampling from the posterior using a single neural function evaluation (NFE), we still need 100 samples to faithfully perform PCA (*i.e.* at least 100 NFEs). This is as opposed to our method, which requires only a single NFE to output the PCs directly. Following your comment, to ensure we highlight this point, we will include in the final manuscript another comparison with MAT (https://arxiv.org/abs/2203.15270) on image inpainting. To the best of our knowledge, MAT is the current SotA in image inpainting on CelebA-HQ (see https://paperswithcode.com/sota/image-inpainting-on-celeba-hq). Our method achieved superior results with RMSE($\downarrow$)/Residual Error Magnitude($\downarrow$) of $10.71/9.42$ for "eyes" and $12.86/11.21$ for "mouth" compared to $11.56/10.66$ and $13.73/12.53$ for MAT, all while being $100\times$ faster.
>
> **The proposed method only provides posterior principle components and does not generate high-quality posterior samples.**
>
> Indeed, as mentioned in the discussion (Section 5), the premise of this work was uncertainty quantification rather than posterior sampling. As you correctly noted, trying to visually interpret the uncertainty structure from dozens of posterior image samples is tedious and heuristic. Therefore, people usually resort to summarizing the samples either to per-pixel variance maps or alternatively to a few principal directions of variation. With this final goal in mind, we designed our method to output the PCs directly. In principle, this weakness could be tackled by employing NPPC in the latent space of a powerful encoder; however, this is beyond the scope of this current work.
>
> **The proposed method is only practical for a small number of principal components which might be insufficient in some applications.**
>
> Indeed, as we acknowledged in the discussion (Section 5), for severely ill-posed inverse problems, a linear subspace with a small number of PCs captures very little of the error. As you correctly point out, we did notice this to be the case for face images on the tasks of super-resolution  and inpainting. Please note, however, that on the task of image colorization presented in the supplementary (also for facial images), a small number of PCs ($K=5$) were actually able to recover large portions of the error. Hence, the practicality of predicting a few PCs to convey posterior uncertainty is eventually dataset and **task** dependent. Nonetheless, beyond a certain number of PCs, navigating the different components may become just as tedious as navigating the original posterior samples.

---

> > ### Comment · Reviewer_2qC5 · 2023-08-21
> >
> > Thanks for your rebuttal.  I think we are in agreement on all points except for one.  I don't feel that it's appropriate to claim that your results are "superior" to MAT because it's not appropriate to judge the quality of inpainting by RMSE.  It's well known that the RMSE metric rewards blurry reconstructions, not sharp/realistic ones, and your "x+w" reconstructions are noticeably blurry.
> >
> > As for the other reviews, I believe their scores are low because they didn't understand some aspects of the research problem (e.g., the lack of ground-truth on which to validate the PCs and variances). That said, in an effort to find a middle ground, I will slightly reduce my overall score from 8 to 7.

---

> > > ### Author Response · Authors · 2023-08-21
> > >
> > > Thanks for the positive score.
> > >
> > > We'd like to clarify a point regarding the RMSE of the prediction. Please note that this RMSE is computed between the GT and the mean prediction of MAT which is also blurry (the average of 100 samples). Of course, we completely agree that RMSE is not a good measure for inpainting. Our goal is only to compare the residual error magnitude, which is the measure that quantifies the accuracy of the estimated PCs. However, it would be unfair to report that number alone, without reporting the RMSE of the prediction, because the error is computed with respect to the prediction. Note that we could report the results of a different experiment, where we train our network to output posterior directions with respect to MAT's mean prediction, rather than with respect to our mean prediction. In this case, we seemingly don't have to report the RMSE of the prediction (as both methods are computed with respect to the same mean). However, in this case our residual error magnitude turns out to be much smaller than that achieved by computing PCA on MAT's predictions. This is because our network has a lot of error to cut from (our first PC captures some of the error that corresponds to the inaccurate mean). We would gladly report these results, but we felt this is a bit deceiving and unfair. This is the reason we chose to report the residual error of each algorithm with respect to its own mean, in which case it seems natural to also report the RMSE of the mean.

---

### Official Review · Reviewer_Dscw · 2023-07-03

**Soundness:** 3 good
**Presentation:** 3 good
**Contribution:** 2 fair
**Rating:** 5
**Confidence:** 2

**Summary:**

This paper proposes using a deep neural network to predict the principle components (PCs), and the associated uncertainties, of the output directly, instead of just the most likely output. This is done by proposing a PC loss that ensures PCs are unit vectors and are orthogonal. The experimental results indicate that the proposed method performs comparably or better than other methods on super-resolution and inpainting tasks, while using much less compute when predicting for new examples.

**Strengths:**

The paper proposes, what is to my knowledge, a novel PC-based method that leads to a more computationally efficient method for super-resolution and inpainting tasks that performs comparable to, or better than, competing techniques. The proposed technique also estimates the data (i.e. aleatoric) uncertainty by learning the variances associated with each PC. Knowing the uncertainty of the predictions could be helpful in understanding the outputs of the proposed method, which is necessary for many scientific applications.

**Weaknesses:**

The main weakness of the paper is evaluations. The proposed method was only compared to two other methods on one dataset (Celeb-A-HQ). This comparison was not done on MNIST or the biological image-to-image translation dataset. The predicted PCs were not compared to the ground truth PCS. Also, the quality of the uncertainty quantification, a key claim of the paper, was not evaluated well.

**Questions:**

- Could the predicted principle components and variances be compared to the ground truths for a dataset such as MNIST? It was done for an extremely simple toy dataset in the supplementary material.

- Is there another way to verify the improvement in uncertainty quantification that the proposed method brings?

- Could the proposed method be compared with other image-to-image translation methods of the biological imaging dataset?

**Limitations:**

The limitation of no guaranteed generalization performance is given. Is there a potential limitation related to choosing the number of PCs to predict? Societal impact was not addressed.

---

> ### Author Rebuttal · Authors · 2023-08-09
>
> **The proposed method was only benchmarked on CelebA-HQ and not on MNIST/biological dataset. Could there be further comparisons on the biological dataset?**
>
> The reason we did not compare to other techniques on the MNIST/biological datasets is that to the best of our knowledge, there exist no pre-trained posterior samplers for these datasets. In all our experiments, we intentionally used only pre-trained models optimized by the respective authors to avoid introducing implementation bias. Specifically, note that for the biological dataset, the authors of [44] used a cGAN based on pix2pix in their implementation, which is known to suffer from mode collapse. Hence, this provided single point estimates for every measurement, preventing us from comparing the PCs. Having said that, in the final manuscript, we will also include further comparisons with MAT (https://arxiv.org/abs/2203.15270) on image inpainting. To the best of our knowledge, MAT is the current SotA in image inpainting on CelebA-HQ (See paperswithcode). Our method achieved superior results with RMSE($\\downarrow$)/Residual Error Magnitude($\\downarrow$) of $10.71/9.42$ for “eyes”' and $12.86/11.21$ for “mouth” compared to $11.56/10.66$ and $13.73/12.53$ for MAT, all while being $100\times$ faster.
>
> **The predicted PCs were not compared to the ground truth (GT) PCs. Could the predicted PCs and variances be compared to the GT on MNIST?**
>
> Thanks for this important question. Please note that there exists no GT uncertainty in image restoration datasets. This is because each element in the dataset is comprised of a **single** posterior sample $\\mathbf{x}\_i\\sim P\_{X\\lvert Y}(\\mathbf{x}\\lvert\\mathbf{y}=\\mathbf{y}\_i)$ for each observed image $\\mathbf{y}\_i$. Therefore, as we stated in the paragraph "The challenge in posterior PCA" (l 153-166), directly computing the posterior covariance $\\mathbb{E}\\left[(\\mathbf{x}-\\hat{\\mathbf{x}})(\\mathbf{x}-\\hat{\\mathbf{x}})^\\top\\lvert\\mathbf{y}\\right]$ (or properties of the covariance such as the top $K$ eigenvectors) is practically impossible. The key implicit assumption underlying our approach (and empirical risk minimization in general) is that the posterior mean $\\mu(\\bf{y})=\\mathbb{E}[\\mathbf{x}\\lvert\\mathbf{y}]$ and the posterior covariance $\\Sigma(\\mathbf{y})=\\mathbb{E}[(\\mathbf{x}-\\mu(\\mathbf{y}))(\\mathbf{x}-\\mu(\mathbf{y}))^\\top\\lvert\\mathbf{y}]$ vary smoothly with $\\mathbf{y}$. Hence, a neural network training on a dataset $\\mathcal{D}=\\left\\{\\left(\\mathbf{x}\_i,\\mathbf{y}\_i\\right)\\right\\}\_{i=1}^{N\_d}$, will be able to capitalize on inter-sample dependencies and learn a smooth approximation of the top $K$ posterior PCs as a function of $\\mathbf{y}$. Having said that, we agree that further verifying the quality of the predicted PCs may be good for reassuring the readers that our method is valid. For this purpose, following common practice (*e.g.* as done in refs [6,25]), we will include a controlled experiment by designing a toy model with a known posterior distribution $P\_{X\\lvert Y}$ (*i.e.* where the PCs are known ground-truth functions of $\\mathbf{y}$). We'll use our approach to train a network to predict the PCs from a fixed dataset of single posterior samples $\\left(\\mathbf{x}\_i \\sim P\_{X\\lvert Y}(\\mathbf{x}\\lvert\\mathbf{y}=\\mathbf{y}\_i), \\mathbf{y}\_i\\right)$, and compare the result to the GT PCs over a test set of $\\mathbf{y}$'s.
>
> **The predicted PCs and variances were not evaluated well. Could the predicted variances be further verified?**
>
> As mentioned above, the quality of the PCs is difficult to ascertain **directly** as there is no GT available. Similarly, for the $k^{\\text{th}}$ predicted variance $\\sigma\_k^2$, we only have the norm of a single projected error $\\lvert\\mathbf{w}\_k^{\\top} \\mathbf{e}\_i\\rvert$ per measurement $\\mathbf{y}\_i$, and hence no GT either. Please note that, unlike per-pixel methods, in our case we cannot compare the aleatoric uncertainty and the test error directly (*e.g.* RMSE vs. fraction of pixels above an uncertainty threshold), because our method does not assume pixels are independent (an incorrect assumption in images). Additionally, please note that we did evaluate the resulting PCs indirectly by measuring the Residual Error Magnitude $\\|\\mathbf{e}-\mathbf{W}\mathbf{W}^{\\top}\\mathbf{e}\\|\_2$ which is a function of the PCs subspace. Nonetheless, in addition to this result, we further verified the predicted variances by comparing the projected test error $\\mathbf{w}\_k^{\\top}\\mathbf{e}\_i$ to the predicted variance $\\sigma\_k^2$ for every test point $\\mathbf{y}\_i$ (See Fig.R1/R2 in the rebuttal PDF). The results indicate that NPPC estimates the standard deviation with high accuracy (mean estimate of 0.96 vs GT of 1). We will add this quantitative validation to our final manuscript.
>
> **Is there a potential limitation related to choosing the number of PCs to predict?**
>
> Thanks, an important point. Theoretically, different measurements may require a different number of PCs $K$ depending on the posterior's complexity. In NPPC, $K$ is a hyper-parameter hard-coded within the network's architecture (it is the number of network outputs) and needs to be set in advance prior to training, acting as an upper bound on the number of possible PCs. This could be computationally inefficient, however, please note that we also predict the standard deviations along each of the $K$ PCs for each input $\\mathbf{y}\_i$. Therefore, if for a certain observation the standard deviations of some of the PCs are small, then the user can simply choose to ignore those PCs. We will touch on this point in our final manuscript.
>
> **Societal impact was not addressed**
>
> We'll comment on this in the final version. In terms of broader impact, proper uncertainty quantification is crucial for trustworthy interpretable systems, particularly in healthcare applications (*e.g.* biological data).

---

### Official Review · Reviewer_aQW2 · 2023-07-09

**Soundness:** 3 good
**Presentation:** 3 good
**Contribution:** 3 good
**Rating:** 6
**Confidence:** 4

**Summary:**

This work proposes to do image recovery inference using posterior principal components. The posterior principal components are often regarded as a function of observed image, but estimating it from a single image is impossible. Therefore, this work proposes to learn the principal components directly by training a neural net on the triplets of data (observed image, ground truth and posterior mean). It learns all the principal components together using shared weights, and at the same time managed to preserve orthogonality. The experimental results show its usefulness in various tasks.

**Strengths:**

1. The paper is very well written and organized
2. The task of the paper is very important and useful
3. Designing and training a neural net that outputs all principal components at once and maintains othorgonality is nontrivial, and the paper solved the problem quite nicely.
4. The experimetal results are encouraging, as the proposed method outperforms the state-of-the-art posterior samplers

Overall, I think it is a solid paper, and its idea is simple and effective

**Weaknesses:**

There is no quality control on the predicted principal component. It is not necessarily a weakness, but it seems to be a common issue for most of deep learning based inference method.

**Questions:**

1. Given different observations, the posterior distribution could be quite different. In this case, do you still select the same number of principal components for each observation?
2. If you train two NNs, where the first NN output 5 components, and the second NN outputs 10 components. Are the top 5 components of the two NNs match each other?

**Limitations:**

yes

---

> ### Author Rebuttal · Authors · 2023-08-09
>
> **There is no quality control on the predicted principal components**
>
> Thanks for this important comment. Please note that as correctly mentioned in your summary, there exists no ground truth uncertainty in image restoration datasets. This is because each element in the dataset is comprised of a **single** posterior sample $\\mathbf{x}\_i\\sim P\_{X\\lvert Y}(\\mathbf{x}\\lvert\\mathbf{y}=\\mathbf{y}\_i)$ for each observed image $\\mathbf{y}\_i$. Therefore, as we stated in the paragraph "The challenge in posterior PCA" (l 153-166), directly computing the posterior covariance $\\mathbb{E}\\left[(\\mathbf{x}-\\hat{\\mathbf{x}})(\\mathbf{x}-\\hat{\\mathbf{x}})^\\top\\lvert\\mathbf{y}\\right]$ (or properties of the covariance such as the top $K$ eigenvectors) is practically impossible. The key implicit assumption underlying our approach (and empirical risk minimization in general) is that the posterior mean $\\mu(\\mathbf{y})=\\mathbb{E}[\\mathbf{x}\\lvert\\mathbf{y}]$ and the posterior covariance $\\Sigma(\\mathbf{y})=\\mathbb{E}[(\\mathbf{x}-\\mu(\\mathbf{y}))(\\mathbf{x}-\\mu(\mathbf{y}))^\\top\\lvert\\mathbf{y}]$ vary smoothly with $\\mathbf{y}$. Hence, a neural network training on a dataset $\\mathcal{D}=\\left\\{\\left(\\mathbf{x}\_i,\\mathbf{y}\_i\\right)\\right\\}\_{i=1}^{N\_d}$, will be able to capitalize on inter-sample dependencies and learn a smooth approximation of the top $K$ posterior PCs as a function of $\\mathbf{y}$. Having said that, we agree that further verifying the quality of the predicted PCs may be good for reassuring the readers that our method is valid. For this purpose, following common practice (*e.g.* as done in refs [6] and [25]), we will include a controlled experiment by designing a toy model with a known posterior distribution $P\_{X\\lvert Y}$ (*i.e.* where the PCs are known ground-truth functions of $\\mathbf{y}$). We'll use our approach to train a network to predict the PCs from a fixed dataset of single posterior samples $\\left(\\mathbf{x}\_i \\sim P\_{X\\lvert Y}(\\mathbf{x}\\lvert\\mathbf{y}=\\mathbf{y}\_i), \\mathbf{y}\_i\\right)$, and compare the result to the ground-truth PCs over a test set of $\\mathbf{y}$'s.
>
> **The complexity of the posterior distribution may vary across observations. Is the number of PCs fixed for every observation?**
>
> Thanks for raising this point. The short answer is yes - *i.e.* the number of predicted PCs $K$ is fixed for different observations. This is because in our approach $K$ is hard-coded within the network's architecture (it is the number of network outputs). We treat $K$ as a hyper-parameter that needs to be set in advance prior to training, acting as an upper bound on the number of possible directions. However, please note that NPPC also predicts the standard deviations along each of the $K$ PCs for each input $\\bf{y}\_i$. Therefore, if for a certain observation the standard deviations of some of the PCs are small, then the user can simply choose to ignore those PCs. We will touch on this point in our discussion.
>
> **If we train two NNs one outputting $K=5$ PCs and one outputting $K=10$ PCs, do the top $K=5$ PCs of both models match?**
>
> That's a great experiment, thanks. We tested this on the Celeba-HQ dataset for the task of image colorization. The resulting first 5 PCs when training two NNs with 5/10 components were very similar (up to a flipped sign) with an average cosine similarity of 0.9 across the first 3 PCs, and  0.83 overall (please see Figures R1/R2 in the rebuttal PDF). For later PCs (*e.g.* the 4th and 5th) the similarity slightly drops as the error variance along multiple PCs is roughly the same, and hence the ordering of the PCs becomes less distinct and prune to optimization errors. To reinforce our results, we also tested the consistency across models outputting $K=1,2,3,4,5$ PCs and found they were also backward consistent. These results further validate that NPPC consistently outputs the PCs in the correct order. We will add this important finding to the appendix and refer to it in our final manuscript.

---

> ### Comment · Reviewer_aQW2 · 2023-08-22
>
> I thank authors for the detailed response. It addressed all my concerns and questions. I raised my score to weak accept.

---

### Author Rebuttal · Authors · 2023-08-09

The PDF includes 2 figures containing the results of experiments proposed by the reviewers.

---

### Decision · Program_Chairs · 2023-09-21

**Decision:**

Accept (poster)

**Comment:**

The initial reviews were mixed. Some issues were identified and were addressed during the rebuttal and discussion period, leading to reviewers to increase their scores.  While one reviewer remained negative, a detailed response was provided by the authors.  I have reviewed both the original review, the authors comments and the paper and believe the concerns have been addressed.  As a result I recommend the paper be accepted. The authors should revise the manuscript to fully address the concerns and confusions raised in the review and discussion process for the final version.